# $d_\chi$-Stencil: A Differential Privacy Mechanism for Interacting with LLMs

## Abstract

The use of language models as remote services requires transmitting private information to external providers, raising significant privacy concerns. This process not only risks exposing sensitive data to untrusted service providers but also leaves it vulnerable to interception by eavesdroppers. Existing privacy-preserving methods for natural language processing (NLP) interactions primarily rely on semantic similarity, overlooking the role of contextual information. In this work, we introduce $d_\chi$-Stencil, a novel token-level privacy-preserving mechanism that integrates contextual and semantic information while ensuring strong privacy guarantees under the $d_\chi$ differential privacy framework, achieving $2\epsilon$-$d_\chi$-privacy. By incorporating both semantic and contextual nuances, $d_\chi$-Stencil achieves a robust balance between privacy and utility. We evaluate $d_\chi$-Stencil using state-of-the-art language models and diverse datasets, achieving comparable and even better trade-off between utility and privacy compared to existing methods. This work highlights the potential of $d_\chi$-Stencil to set a new standard for privacy-preserving NLP in modern, high-risk applications. Our code is available at:
`https://anonymous.4open.science/r/Dchistencil-FFF1/README.md`.

## 1 Introduction

Natural language processing (NLP) models as a service, such as ChatGPT OpenAI (2021); Ouyang et al. (2022), present notable privacy challenges. These models often require users to send their data to external servers, which can result in potential exposure, misuse, or insufficient protection of personal and sensitive information Sousa & Kern (2023). For example, information can be leaked by model inversion Li et al. (2017); Devlin et al. (2019), exploitation of feature memorization within the large language model (LLM) Carlini et al. (2021), and more. This issue affects not only users but also service providers, who are required to adhere to regulations such as the General Data Protection Regulation (GDPR) Voigt & Von dem Bussche (2017) and the California Consumer Privacy Act (CCPA) Barrett (2019). Moreover, the dependence on third-party services further complicates compliance, as users must rely on these providers to ensure that they follow these privacy standards and manage data responsibly.

One common approach to addressing privacy concerns in NLP is the application of differential privacy mechanisms Dwork (2006), which offer formal guarantees by limiting the amount of information that can be inferred about any individual user. These mechanisms can be employed during either the training or inference phases of an LLM, typically by perturbing the input data with carefully calibrated noise. This process effectively conceals the original input, ensuring that private details cannot be reconstructed from the model's outputs. Differential privacy can operate in both remote and local settings. However, given the inherent trust issues with remote servers, a more practical and secure alternative is to apply differential privacy locally (LDP; Arachchige et al., 2019). In this local setting, users preprocess their data using differential privacy techniques before sharing sanitized results with a server. This approach ensures that raw data remains protected from potential breaches, significantly reducing privacy risks while still enabling useful contributions to the model's functionality.

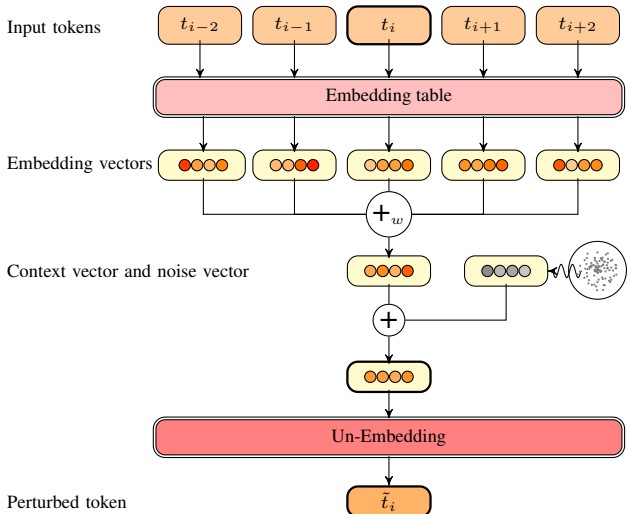

Figure 1: Schematic overview of our proposed method, $d_\chi$-STENCIL, which integrates contextual and semantic information to enhance privacy preservation while maintaining the utility of the NLP models.

In this work, we introduce $d_\chi$-STENCIL, a novel differential privacy-based technique for token-level privacy preservation. The core idea of $d_\chi$-STENCIL is to integrate contextual and semantic information while ensuring strong privacy guarantees that comply with LDP, as seen in Figure 1. Specifically, for each token in a sentence, we encode the embedding vectors of its nearest neighbors to capture contextual relationships. To further enhance privacy, we introduce a noise vector sampled from a random distribution, which perturbs the encoded vector and makes it harder to reconstruct. We show that for a noise vector sampled from a Laplacian distribution, and some privacy parameter $\epsilon$, the mechanism is $2\epsilon$-$d_\chi$-private. Finally, we locate the closest token (the process of un-embedding an embedding vector) to the perturbed vector, thereby preserving semantic integrity. By embedding information from neighboring tokens directly into each token, $d_\chi$-STENCIL effectively maintains contextual information, an aspect critical to the success of LLMs but often overlooked by existing privacy-preserving techniques that primarily focus on semantic similarity. This approach offers the potential for improved utility compared to such techniques, striking a more effective balance between privacy and utility.

To evaluate the efficacy of the proposed privacy-preserving mechanism, we conduct a comprehensive set of experiments on novel benchmarks designed to assess the full capabilities of LLMs, for instance, the SWAG Zellers et al. (2018b) benchmark. We compare the $d_\chi$-STENCIL mechanism against existing privacy-preserving techniques, including the CUSTEXT$^+$ Chen et al. (2023) text sanitization method and the $d_\chi$-differential privacy approach proposed by Feyisetan et al. (2020). The experimental results show that the $d_\chi$-STENCIL mechanism achieves comparable and even better utility-privacy tradeoffs compared to these alternative methods, demonstrating the potential of incorporating contextual information in addition to semantic information in privacy preserving techniques.

## 2 PRESERVING PRIVACY IN MODELS

There have been many attempts to preserve privacy which primarily focus on anonymization techniques Liu et al. (2017); Friedrich et al. (2019) and methods that introduce noise into embedding vectors Zhou et al. (2023; 2022). Although these approaches provide some degree of privacy protection, they often lack rigorous mechanisms to quantify the guarantees they offer, which may make them insufficient for compliance with strict privacy regulations. By contrast, the LDP mechanism offers a robust framework with formal guarantees, making it a reliable and regulation-compliant approach for safeguarding privacy in NLP applications Cummings & Desai (2018).

Since the primary goal is to maintain the usefulness of the NLP model, the privacy-preserving technique must achieve a good balance between privacy and utility. In this regard, the LDP mechanism imposes a high privacy standard, requiring that any two samples produce similar and indistinguishable output distributions Feyisetan et al. (2020). This often results in outputs that lack sufficient information, thereby harming downstream tasks. To address this issue, a common practice is to implement a relaxed version of LDP known as $d_\chi$-privacy Feyisetan et al. (2020); Qu et al. (2021); Yiwen et al. (2018). $d_\chi$-privacy requires the output distribution to be proportionate with respect to the distance between any two samples. Consequently, the outputs are likely to preserve more semantic information as related by the distance function.

Qu et al. (2021) proposed a $d_\chi$-LDP mechanism that incorporates noise from a random distribution into the user's input. The noise introduced complies with formal privacy guarantees under the privacy parameter $\epsilon$, which determines the amount of noise introduced to the embedding vector. They outline three distinct methods for noise application across LLM components: the input text, token embeddings, and sequence embeddings. Although applying noise at the sequence embedding level has the least impact on performance, it requires access to the remote LLM, which may be restricted due to proprietary limitations. While this approach is straightforward to implement and ensures privacy, it demands a high privacy parameter for effective protection. This may inadvertently facilitate token reconstruction, even by relatively simple adversaries Harel et al. (2024).

As an alternative to this noise-based mechanism, SanText Yue et al. (2021) employs an exponential variant of LDP that utilizes random sampling from a list of tokens McSherry & Talwar (2007). The approach computes a distance metric between each input token and all other tokens in the vocabulary, typically using euclidean distance in the embedding space to capture semantic relationships. Using these distances and a privacy parameter $\epsilon$, SanText employs an exponential sampling mechanism to replace original tokens: given a value of $\epsilon$, the probability of selecting each replacement token is proportional to $e^{-\epsilon \, \mathrm{dist} /2}$, where $\mathrm{dist}$ is the distance to the original token. Thus, tokens with smaller distances (higher semantic similarity) to the original have a higher probability of being selected as replacements. This approach offers a better trade-off between performance and privacy compared to Qu et al. (2021)'s method.

While innovative, SanText's sampling procedure has a key limitation: since replacement tokens are sampled from the entire vocabulary, there remains a non-negligible probability of selecting tokens that are semantically dissimilar to the original, potentially degrading utility without meaningful privacy benefits. To address this limitation, CUSTEXT$^+$ Chen et al. (2023) introduces a refined approach. Instead of sampling from the complete token set, it first generates a candidate pool of the $K$ most semantically-similar tokens to the original. The mechanism then applies SanText's exponential sampling procedure to this restricted set. By constraining the sampling space to semantically-related tokens, CUSTEXT$^+$ achieves superior utility even at lower values of the privacy parameter $\epsilon$, as it guarantees replacements that better preserve the original token's meaning.

All these methods provide valuable privacy guarantees, but several important directions remain unexplored. For example, the potential benefits of incorporating contextual information, the applicability to generative LLMs, and evaluation under practical attack scenarios are yet to be investigated. Of these, the role of **context** appears to be most crucial, given its centrality in textual language data and fundamental duty in LLMs. Harel et al. (2024) introduced a privacy-preserving mechanism leveraging both contextual and semantic information to enhance LLM utility. The STENCIL mechanism encodes each token's embedding vector alongside its neighboring tokens' vectors, effectively preserving contextual nuances. However, the absence of a random component constrains its privacy guarantees, limiting the mechanism's comprehensive privacy protection.

## 3 CONTEXT- AND SEMANTIC-BASED PRIVACY PRESERVATION

One method commonly employed for ensuring local differential privacy involves injecting a controlled amount of *noise* into various model components, thereby obfuscating the original input. These components may include sequence embeddings, token embeddings, or the tokens themselves Mosallanezhad et al. (2019); Feyisetan et al. (2020); Lyu et al. (2020); Qu et al. (2021); Zhou et al. (2022). The success of NLP models in most tasks, however, is primarily attributed to their ability to leverage contextual information. Thus, privacy-preserving mechanisms that incorporate contextual

information have the potential to achieve better downstream task performance compared to those that do not consider context.

We introduce $d_\chi$-STENCIL, a privacy preservation technique based on the STENCIL mechanism and the $d_\chi$ noise mechanism. Like STENCIL, $d_\chi$-STENCIL encodes contextual information from neighboring tokens. Incorporating the $d_\chi$ noise component enhances privacy protection and ensures compliance with LDP and privacy regulations.

### 3.1 THE STENCIL MECHANISM

The STENCIL Harel et al. (2024) privacy mechanism operates by replacing each token $t_i, \forall i \in \{0 \ldots N\}$ (where $N$ is the number of tokens in the input) with a new token, $t_i \to \tilde{t}_i$, that aims to preserve both semantics and context. To derive the new token $\tilde{t}_i$, information from its $L$ neighboring tokens $t_k, \forall k \in \{i - \frac{L}{2}, \ldots, i + \frac{L}{2}\}$ is incorporated. This is achieved by obtaining the embedding vector representations, $\phi_k$, of each of the $L + 1$ tokens using an embedding lookup table, $\mathbf{E} \in \mathbb{R}^{|\mathcal{V}| \times ||\phi||}$, where $|\mathcal{V}|$ is the size of the vocabulary and $||\phi||$ is the dimension of the vector. Next, the embedding vectors, $\phi_k = \mathbf{E}[t_k]$, are combined using a normalized weighted function $f_k$ (satisfying $\sum_{k=i-\frac{L}{2}}^{i+\frac{L}{2}} f_k = 1$) to generate a new embedded quasi-vector $\tilde{\phi}_i$. The new perturbed token $\tilde{t}_i$ is then selected as the token from the vocabulary $\mathcal{V}$ with the smallest distance to $\tilde{\phi}_i$ that is different than the original one, i.e., $\arg\min_{t_j \in \mathcal{V}} \text{dist}(\mathbf{E}[t_j], \tilde{\phi}_i)$, where dist is a distance scoring function.

Although the STENCIL mechanism obscures the original token and offers a certain level of privacy, it does not adhere to the principles of differential privacy due to the absence of a randomized component. This deterministic nature not only limits its compliance with formal privacy guarantees but also renders it highly vulnerable to reconstruction attacks, as adversaries can exploit the lack of randomness to reverse-engineer the original data.

### 3.2 THE $d_\chi$ MECHANISM

The relaxed $d_\chi$ mechanism, or the NOISE mechanism, proposed by Feyisetan et al. (2020) and Qu et al. (2021) requires that for any two inputs $x, x' \in X$ with a randomized function $M : X \to X$, the following holds:

$$\frac{Pr[M(x) = y]}{Pr[M(x') = y]} = e^{\eta \, \text{dist}(x,x')}, y \in X, \tag{1}$$

where $\text{dist}(x, x')$ is a distance function, such as cosine similarity or euclidean distance, and $\eta$ is a privacy parameter. This condition guarantees that the probability distributions over outputs for similar inputs are close. For instance, if the distance function is defined over the embedding space, smaller distances can correlate to semantic similarity.

To apply the $d_\chi$ mechanism in LLMs, Qu et al. (2021) proposed that the randomized function be defined as $M(x) = x + \mathbf{p}$, where $x \in \mathbb{R}^{||\phi||}$ represents the sequence embedding vector representation, and $\mathbf{p} \in \mathbb{R}^{||\phi||}$ is a noise vector sampled from a distribution whose density is proportionate to $\eta$. Specifically, we generate $\mathbf{p}$ by multiplying a sample from a Gamma distribution $\Gamma(||\phi||, 1/\eta) = \frac{r^{||\phi||-1} \eta^{||\phi||} e^{-r\eta}}{(||\phi||-1)!}$ and a uniform sample from a unit hypersphere ($\frac{\Gamma(||\phi||/2)}{||\phi|| \pi^{||\phi||/2}}$).

For the token-level implementation of this mechanism, we introduce pre-processing and post-processing stages. The pre-processing step involves obtaining the embedding vector representation of the token $t$ from an embedding lookup table $\mathbf{E}[t]$. Then, the $d_\chi$ mechanism is applied on $\mathbf{E}[t]$ as described. Finally, the post-processing step returns the token $t'$ that is closest (according to the distance function $d$) to the output of the $d_\chi$ mechanism, effectively yielding a noised but semantically similar token.

### 3.3 THE $d_\chi$-STENCIL MECHANISM

Our proposed method, $d_\chi$-STENCIL, integrates the $d_\chi$ mechanism with the STENCIL approach, thereby maintaining privacy guarantees while leveraging the contextual and semantic information of the text to better preserve utility. The $d_\chi$-STENCIL mechanism follows the same procedure as the

STENCIL mechanism, with one key difference. After computing the quasi-embedding vector $\tilde{\phi}_i$, a calibrated noise $\mathbf{p}$, derived as explained in §3.2, is added. The process is described in Algorithm 1.

---

**Algorithm 1** Overview of the STENCIL Mechanism.

---

    **Input**: Embeddings table $\mathbf{E}$, $N$ tokens, $\sigma$, $L$
  **for** $i = 0 \ldots N$ **do**
    $\tilde{\phi}_i \leftarrow 0$
    **for** $j = \min\left(0, i - \frac{L}{2}\right) \ldots \max\left(i + \frac{L}{2}, N\right)$ **do**
      $\phi_j \leftarrow \mathbf{E}[t_i]$
      $f(j, \sigma) \leftarrow e^{-j^2/2\sigma^2}$
      $\tilde{\phi}_i \leftarrow \tilde{\phi}_i + f(j, \sigma) \cdot \phi_j$
    **end for**
    $\tilde{\phi}_i \leftarrow \tilde{\phi}_i / \sum_j f(j, \sigma) + \mathbf{p}$
    $\tilde{t}_i \leftarrow \arg\min_{t_j \in \mathcal{V}} \mathrm{dist}\left(\mathbf{E}[t_j], \tilde{\phi}_i\right)$
  **end for**

---

The privacy proof is supplied in the Appendix A.

The privacy-utility trade-off of the $d_\chi$-STENCIL method can be controlled by adjusting the window size $L$, the properties of the weighted function $f$, and $\eta$. In our study, we employ a Gaussian smoothing function as the weighting function, where the standard deviation $\sigma$ plays a pivotal role in balancing privacy and downstream task performance. For odd-numbered window size, the Gaussian weights concentrate on the original token, which may enhance accuracy but also lower privacy. In contrast, an even-numbered window size, the highest weights are distributed between the original token and its nearest neighbor, resulting in a more balanced allocation. In §5, we analyze the impact of these parameters on accuracy and privacy.

## 4 EXPERIMENTS

We evaluate the impact of $d_\chi$-STENCIL using both odd- and even-numbered window sizes on downstream task performance, comparing its effectiveness against CUSTEXT$^+$, STENCIL, and the NOISE. To provide a comprehensive comparison, we examine the different privacy preserving techniques on two distinct LLMs: Google's T5-Flan large variant (FLAN-T5; Chung et al., 2022)[1] and Qwen2.5-1.5B-Instruct (QWEN2.5; Yang et al., 2024)[2]. The models were used without fine-tuning on a specific task. The embedding lookup table utilized in this study is GloVe Pennington et al. (2014),[3] which contains 2.2 million tokens and was trained on a corpus of 840 billion tokens.

For all methods, the distance score $\mathrm{dist}$ was calculated using cosine similarity, i.e., $\mathrm{dist} = \frac{\mathbf{E}[t_j] \cdot \tilde{\phi}_i}{\|\mathbf{E}[t_j]\| \cdot \|\tilde{\phi}_i\|}$. All experiments were conducted on a server with two AMD EPYC 7763 64-Core processors and an NVIDIA RTX 6000.

**Stopword exclusion** Identifying sensitive words, such as those involved in named entity recognition (e.g., names, addresses, workplaces), is a challenging task typically approached using statistical methods Liu et al. (2017); Cohn et al. (2019); Poostchi et al. (2018); Friedrich et al. (2019). As a result, our mechanism treats all tokens as sensitive, since we cannot reliably distinguish sensitive from non-sensitive tokens. However, since treating stopwords as non-sensitive may pose a low privacy risk Chen et al. (2023), we apply all the privacy preserving mechanisms to all words except stopwords.

### 4.1 DATASETS

To evaluate the impact of various privacy-preserving techniques on the performance of generative LLMs, we select several benchmark datasets that assess a wide range of model capabilities. By

---

[1]https://huggingface.co/google/flan-t5-large
[2]https://huggingface.co/Qwen/Qwen2.5-1.5B-Instruct
[3]https://nlp.stanford.edu/projects/glove/

testing our techniques on these well-established datasets, we ensure that our privacy interventions are assessed in terms of their real-world effectiveness, offering a clear understanding of how they affect model performance on standard NLP tasks. The benchmarks that were selected are included in the LM evaluation harness Gao et al. (2023): SST2 Socher et al. (2013), QNLI Wang et al. (2018), SWAG Zellers et al. (2018a), and MMLU Hendrycks et al. (2020).

Each privacy-preserving technique was applied to the test set of the datasets, creating a privatized test set. The resulting privatized test set was subsequently used to evaluate both utility and privacy, without any model training on the original or privatized data.

During the experiments we observed that some tokens were out of vocabulary (OOV) i.e., were not included in the embeddings table of GloVe. For these OOV tokens, their original forms were retained, which improved model accuracy but also lowered privacy. Nevertheless, the amount of OOVs for all datasets was lower than 10% and all privacy-preserving techniques were similarly affected, ensuring that the comparisons remain valid and fair.

## 4.2 NEAREST-NEIGHBOR RECONSTRUCTION

The core principle of the privacy-preserving mechanisms introduced, namely CUSTEXT$^+$, STENCIL, $d_\chi$-STENCIL, and NOISE, is to maintain semantic relationships to minimize their impact on the LLM's performance. This is achieved by substituting tokens with semantically similar alternatives, determined by the embedding space as explained: $\arg\min_{t_j \in \mathcal{V}} \text{dist}(\mathbf{E}[t_j], \tilde{\phi}_i)$. However, this strategy may be vulnerable to adversarial exploitation, where an attacker attempts to reverse-engineer the substitution process under the assumption that the attacker has access to embeddings table of the privacy mechanism. Specifically, given a perturbed token $t'$, the attacker can extract its embedding vector representation $\mathbf{E}[t']$. By calculating the cosine similarity or euclidean distance between this perturbed embedding and the embeddings of other tokens in the vocabulary ($\mathbf{E}[t]$ for $t \in \mathcal{V}$), the attacker can identify candidate original tokens by ranking them according to their similarity scores and selecting those above a chosen threshold (for example, the top 5 candidates), potentially revealing the original token with high probability.

To evaluate the robustness of these techniques against token inversion attacks, we implemented this attacker model and assessed whether any original token appeared among the top five nearest neighbors of the perturbed token. Finally, we report the likelihood as the reconstruction rate Pr@5.

## 4.3 RESULTS

In Figure 2, we present comparative accuracy-privacy trade-offs across multiple privacy-preserving mechanisms using FLAN-T5 on the SST2 and QNLI datasets, and QWEN2.5 on the SWAG and MMLU datasets. The charts are constructed such that better models are closer to the top left corner. For $d_\chi$-STENCIL, we demonstrate results with both odd and even-numbered window sizes $L$ using optimal $\sigma$ values across varying privacy parameter $\eta$. We include STENCIL results with optimal parameters ($L = 5$, $\sigma = 1.25$), alongside NOISE and CUSTEXT$^+$ with varying values of the privacy parameters $\eta$ and $\epsilon$, respectively.

Across all experimental configurations, $d_\chi$-STENCIL with even-numbered $L$ demonstrates superior utility-privacy tradeoffs, achieving optimal accuracy while maintaining low reconstruction rates. However, this configuration exhibits an inherent accuracy ceiling even at high $\eta$ values, making it suboptimal for applications prioritizing maximum utility. The relationship between window size parity and performance characteristics is examined in detail in §5.

Moreover, when excluding $d_\chi$-STENCIL with even-numbered $L$, the SWAG dataset exhibits minimal variance across mechanisms, likely attributable to its lower average token count. This suggests that the cumulative noise effect, which scales with sentence length, remains comparatively low for SWAG relative to other datasets in our evaluation.

The odd-number $L$ of $d_\chi$-STENCIL mechanism demonstrates better utility and privacy compared to STENCIL and NOISE mechanisms for all cases. By integrating a noise component, $d_\chi$-STENCIL enhances privacy beyond STENCIL, while preserving information more effectively than NOISE, thus representing an improvement for both privacy-preserving techniques.

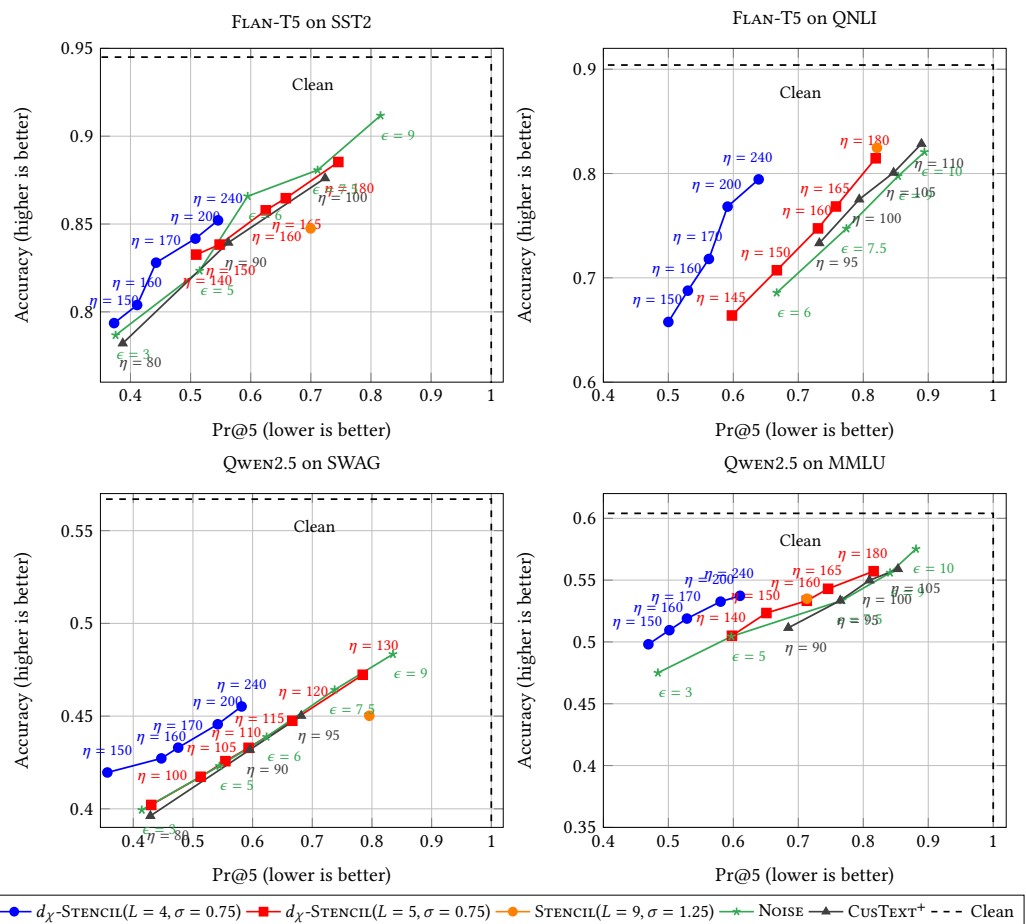

Figure 2: Comparison of optimal accuracy and reconstruction rates (Pr@5) across privacy-preserving mechanisms: $d_\chi$-STENCIL (varying $\eta$, $L$), STENCIL, CUSTEXT$^+$ (varying $\epsilon$), and NOISE (varying $\eta$). Results shown for: (top row) FLAN-T5 on SST2 and QNLI datasets; (bottom row) QWEN2.5 on SWAG and MMLU datasets. Clean baseline represents unsanitized data performance.

Our results show that contextual information has varying effects across different benchmarks, with stronger benefits observed in QNLI and MMLU compared to SST2 and SWAG datasets. This suggests that the effectiveness of using contextual information in privacy mechanisms may depend on the specific task being performed.

Nevertheless, across all evaluated datasets, $d_\chi$-STENCIL demonstrates comparable or even better utility-privacy tradeoffs, validating the effectiveness of incorporating contextual information into privacy-preserving mechanisms. These results suggest that context-aware approaches can enhance the fundamental trade-off between utility preservation and privacy protection in language model applications.

## 5 PARAMETER STUDY

To investigate the impact of window size $L$ and the standard deviation $\sigma$ of the gaussian weights $f_i$ on the accuracy and resilience against reconstruction attacks, we conducted tuning experiments for these parameters. We note that although these results are demonstrated using QWEN2.5 on the SST2 and SWAG datasets, they are consistent for other models and datasets.

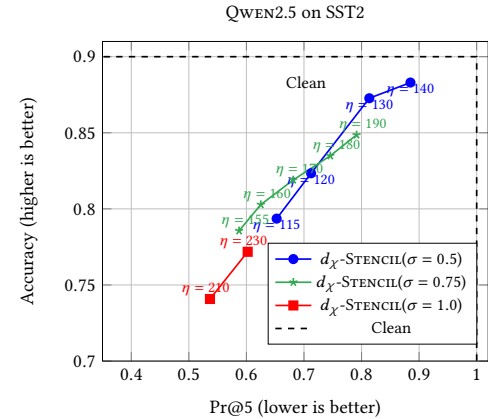

Figure 3: Accuracy-privacy utility comparison of $\sigma = 0.5, 0.75, 1.0$ with varying $\eta$ values and a fixed size of $L = 9$ using QWEN2.5 performance on the SST2 dataset. The Clean trend represents the unsanitized baseline.

## 5.1 IMPACT OF $\sigma$

In $d_\chi$-STENCIL and STENCIL, lower $\sigma$ values assign higher weights to the original token, rendering the perturbed token more similar to the original and reducing encoded contextual information. Since contextual information can help preserve utility, we empirically investigate its impact on the utility-privacy tradeoff by varying $\sigma$ (fixed $L = 9$) on the SST2 dataset using QWEN2.5 in Figure 3.

At $\sigma = 0.5$, a sharp utility rise occurs with increasing $\eta$ because smaller noise vectors and higher original token weights increase the probability of token preservation. While optimal for accuracy, this scenario compromises reconstruction resistance.

At $\sigma = 0.75$, the introduction of additional contextual information necessitates higher values of $\eta$ compared to $\sigma = 0.5$. Notably, the results reveal that for reconstruction rates below approximately 0.7, $\sigma = 0.75$ consistently achieves better utility than $\sigma = 0.5$. This finding underscores the potential of embedding contextual information within the privacy mechanism to enhance both utility and privacy. However, the poor performance observed at $\sigma = 1.0$ suggests that incorporating contextual information requires a more refined and strategic approach to maintain a favorable utility-privacy balance.

## 5.2 IMPACT OF WINDOW SIZE $L$

When $L$ is an odd number, the gaussian weights distribute such that the largest weight is assigned to the original token, potentially increasing accuracy but also raising the reconstruction risk. Conversely, with an even-numbered $L$, the highest gaussian weights spread across the original token and its nearest neighbor, yielding a more distributed weight allocation. This weight distribution suggests that tokens replaced with an even-numbered neighborhood size may be less similar to the original, potentially decreasing accuracy while simultaneously reducing reconstruction vulnerability. We empirically validate these observations by comparing performance-privacy metrics for even and odd neighborhood sizes across different $\eta$ values, ensuring matched reconstruction rates, using QWEN2.5 on the SWAG dataset with $\sigma = 0.75$ in Figure 4.

The results show that even-numbered $L$ exhibits limited accuracy, a characteristic persisting even with negligible vector noise (increasing $\eta$), as the perturbed token is not guaranteed to match the original one. Despite this accuracy constraint, up to its performance ceiling, even-numbered $L$ demonstrates superior performance compared to odd-numbered $L$, delivering higher accuracy while simultaneously maintaining a lower reconstruction rate. Nevertheless, the odd-numbered $L$ is preferable in scenarios prioritizing high utility.

Altering the window size for odd-numbered $L$ marginally impacts utility and privacy, whereas even-numbered $L$ shows a slight preference for lower window sizes.

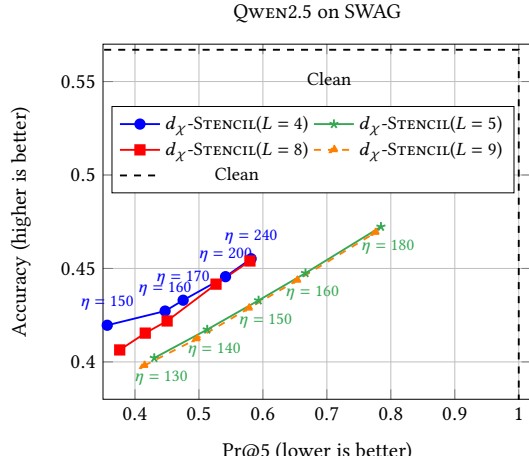

Figure 4: Accuracy-privacy utility comparison of odd and even neighbor counts $L$ for varying $\eta$ values, with fixed $\sigma = 0.75$ using QWEN2.5 performance on the SWAG dataset. The Clean trend represents the unsanitized baseline.

Overall, the results of varying $\sigma$ and $L$ demonstrate that careful parameter tuning can lead to high utility while also optimizing the trade-off between utility and privacy. More importantly, these findings highlight that an improved balance between utility and privacy can be achieved, particularly when leveraging the contextual component of $d_\chi$-STENCIL.

## 6 CONCLUSION

In this paper, we introduced $d_\chi$-STENCIL, a novel $d_\chi$ privacy-preserving technique that utilizes both semantic and contextual information in order to maximize utility while safeguarding individual privacy. In order to evaluate the utility-privacy tradeoff caused by this mechanism, we evaluated it comparing to three other privacy preserving mechanisms: STENCIL, NOISE and CUSTEXT$^+$. Because LLMs are widespread mainly in the form of chatbots, we conducted these experiments over standard benchmarks such as SST2 and QNLI, and SWAG and MMLU. Our results demonstrate that incorporating both contextual and semantic information can provide a better utility-privacy trade-off compared to methods that rely solely on semantic information. In addition, the results indicate that more sophisticated methods incorporating contextualized information can yield even better results.

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

## A  DIFFERENTIAL PRIVACY OF $d_\chi$-STENCIL

In this section, we prove that $d_\chi$-STENCIL is $d_\chi$-private, with the main result stated in Theorem 1. As preliminary steps preceding the theorem, we define the distance functions for which the theorem applies, state the noise distribution $\mathbf{p_i}$ in more convenient form and prove Lemma 1, which is a bound on the overall contribution of a specific token to the output.

We prove Theorem 1 for two distance functions defined as follows. Consider two vectors $x = (x_1, \ldots, x_N), (x'_1, \ldots, x'_N) \in \left(\mathbb{R}^\ell\right)^N$ and define the first distance function as the sum of the Euclidean distances between the components, i.e.,

$$d_2(x, x') = \sum_{i=1}^{N} ||x_i - x'_i||,$$

where $||z_i|| = \sqrt{z_{i,1}^2 + \ldots + z_{i,\ell}^2}$ is the Euclidean norm for any $z_i = (z_{i,1}, \ldots, z_{i,\ell}) \in \mathbb{R}^\ell$. The second distance function is defined as the sum of the cosine distances between the components, i.e.,

$$d_C(x, x') = \sum_{i=1}^{N} 1 - \frac{\langle x_i, x_i' \rangle}{||x_i|| \cdot ||x_i'||},$$

where $\langle \cdot, \cdot \rangle$ denotes the inner product in $\mathbb{R}^\ell$ and $\frac{\langle x_i, x_i' \rangle}{||x_i|| \cdot ||x_i'||}$ is cosine similarity.

We define the perturbation $\mathbf{p_i}$ in the algorithm using the Laplacian density function in $\ell$ dimensions from Fernandes et al. (2019). The function with parameter $\epsilon > 0$, is defined for every vector $v = (v_1, \ldots, v_\ell) \in \mathbb{R}^\ell$ by $\mathrm{Lap}_{\ell,\epsilon}(v) = ce^{-\epsilon||v||}$, where $||v||$ is the Euclidean norm and $c = c(\ell, \epsilon)$ is chosen so that the associated cumulative distribution function is correct. That is, it is chosen so that $\int_{-\infty}^{\infty} \ldots \int_{-\infty}^{\infty} ce^{-\epsilon||v||} dv_1 \ldots dv_\ell = 1$.

Fernandes et al. (2019) prove that the noise sampled in Algorithm 1 as a product of a gamma distribution and a uniform distribution on specific domains is distributed with the Laplacian density function in $\ell$ dimensions if $\mathbf{E}(t_i) \in \mathbb{R}^\ell$ for every token $t$. In the $d_\chi$-STENCIL mechanism, each token $t_i$ is associated with $\tilde{\phi}_i$, a weighted sum of the embeddings of all tokens $t_j$ within a window of at most $L$ tokens around $t_i$. Denote the weights used to compute $\tilde{\phi}_i$ by $f'_{i,j}$ before normalization and by $f_{i,j} = f'_{i,j} / \sum_j f'_{i,j}$ after normalization, which implies that $\sum_j f_{i,j} = 1$. The specific weights $f'_{i,j}$ used in the algorithm have two important properties: *symmetry* around $i$, meaning that $f'_{i,i-j} = f'_{i,i+j}$ for all $i, j$, and *decay* with distance from $i$, i.e. $f'_{i,j'} \leq f'_{i,j}$ for all $i$, and all $j, j'$ such that $|i - j| \leq |i - j'|$.

Now, instead of considering the contribution of all neighboring tokens $t_j$ to $\tilde{\phi}_i$, we estimate the total contribution of a token $t_j$ to the $\tilde{\phi}_i$ of all its neighbors. Let $C_j = \sum_{i=1}^{N} f_{i,j}$ be the *contribution* of a token $t_j$. The following lemma bounds $C_j$.

**Lemma 1.** *For any non-negative integers $N, L$ the $d_\chi$-STENCIL mechanism satisfies the two following properties:*

- $C_j = 1$ *for all $j$ such that $L \leq j \leq N - L$.*

- $C_j \leq 2$ *for all $j$ such that $0 \leq j \leq N$.*

*Proof.* If $L \leq j \leq N - L$, then $t_j$ contributes to values $\tilde{\phi}_i$ in the range $L/2 \leq j - L/2 \leq i \leq j + L/2 \leq N - L/2$. The window for each of the associated tokens $t_i$ is of length $L$, and therefore the sequence of weights $(f_{i,i-L/2}, \ldots, f_{i,i+L/2})$ is identical for all these tokens, including $t_j$. It follows that due to the symmetry property noted above, the contribution of $t_i$ to $\tilde{\phi}_j$ is equal to the contribution of $t_j$ to $\tilde{\phi}_i$, or formally $f_{i,j} = f_{j,i}$. Therefore, $C_j = \sum_{i=0}^{N} f_{i,j} = \sum_{i=0}^{N} f_{j,i} = 1$, which proves the first part of the lemma.

For general $j$, $0 \leq j \leq N$, let $L'$ (respectively $L''$) be the number of tokens to the left (resp. right) of $t_j$ that contribute to $\tilde{\phi}_j$, and assume w.l.o.g. that $L' \leq L'' \leq L/2$. Let $F_i = \sum_j f'_{i,j}$ be the normalization factor for the $i$-th token, that is, $f_{i,j} = f'_{i,j}/F_i$ for all $i, j$. By the symmetry of the weights we have that $f'_{i,j} = f'_{j,i}$. Note that symmetry implies that at least half the weight $F_j$ is concentrated in $f'_{j,j}$ and to its right, that is $f'_{j,j} + \sum_{i=1}^{L''} f'_{j+i,j} \geq F_j/2$. To prove the second part of the lemma's statement, we show that $F_i \geq F_j/2$ for all $j' - L' \leq i \leq j' + L''$.

We start with $t_{j-L'}$, which is the leftmost token that receives contribution from $t_j$. That token receives contribution from at least $L''$ tokens to the right of it, since it is not to the right of $t_j$. Therefore, $F_{j'-L'} \geq f'_{j,j} + \sum_{i=1}^{L''} f'_{j+i,j} \geq F_j/2$. We now move right from $t_{j-L'}$ maintaining for each token a set of $L'' + 1$ tokens that contribute to it. For $t_i$ in the range $j - L' \leq i \leq j - L' + L''$ that set is exactly $\{t_{j-L'}, \ldots, t_{j-L'+L''}\}$, while in the range $j - L' + L'' < i \leq j + L''$ that set is $\{t_{i-L''}, \ldots, t_i\}$. In the second range, due to symmetry, $F_i = F_{j-L'} \geq F_j/2$. In the first range, due to both symmetry and decay with distance from the token, we have that $F_i \geq F_{j-L''} \geq F_j/2$.

Therefore, $C_j = \sum_{i=0}^{N} f_{i,j} = \sum_{i=0}^{N} f'_{i,j}/F_i \le \sum_{i=0}^{N} 2f'_{j,i}/F_j = 2$, which completes the proof of the lemma.

$\square$

The following theorem states the differential privacy property that the $d_\chi$-STENCIL mechanism satisfies. Its proof depends on the Laplace distribution of the noise.

**Theorem 1.** *For all non-negative integers $\ell, L, N$, for all embedding functions $\mathbf{E}$ mapping tokens to $\mathbb{R}^\ell$, and for every $\epsilon > 0$, the $d_\chi$-STENCIL mechanism with parameters $L, N, \mathbf{E}$ and Laplace noise with parameter $\epsilon$ is $2\epsilon$-$d_\chi$-private for both the $d_2$ and the $d_C$ distance functions.*

*Proof.* Let $t = (t_1, \ldots, t_N), t' = (t'_1, \ldots, t'_N)$ be two sequences of $N$ input tokens and let $x = \mathbf{E}(t) = (x_1, \ldots, x_N), x' = \mathbf{E}(t') = (x'_1, \ldots, x'_N)$ be the associated sequences of embeddings in $\mathbb{R}^\ell$. Let $M(x)$ denote the output of $d_\chi$-STENCIL on a sequence of embeddings $x$. If $y = M(x)$, then $y = (y_1, \ldots, y_N)$ is a sequence of $N$ output embeddings.

The proof proceeds in three steps. The first step assumes that $d_\chi$-STENCIL uses Euclidean distance and gives a bound on the probability that the $i$-th output component is $y_i$ as a function of the weighted distance between the components of $x$ and $x'$ around $i$. The second uses the independence of the $\mathbf{p_i}$ noise components to prove that $d_\chi$-STENCIL is $2\epsilon$-$d_\chi$-private for Euclidean distance between the embeddings. The third step proves the theorem for cosine distance.

For every $1 \le i \le N$, let $A_{y_i} \subseteq \mathbb{R}^\ell$ be the subset of all vectors that are closer to $y_i$ than to $z$ for any possible output $z$. $d_\chi$-STENCIL with input $t$ returns output $y$ if and only if for every $i = 1, \ldots, N$, the weighted sum of the embeddings in $x$ together with the $i$-th noise component $\mathbf{p_i}$ is in $A_{y_i}$. Let $r_i + \mathbf{p_i} = \sum_j f_{i,j} x_j + \mathbf{p_i}$ be the random variable which measures the probability that on input $t$ the mechanism outputs $y_i$ in its $i$-th component. Similarly $r'_i + \mathbf{p_i} = \sum_j f_{i,j} x'_j + \mathbf{p_i}$ measures the same probability when the input is $t'$. The probability that $r_i + \mathbf{p_i} \in A_{y_i}$ is

$$\Pr[r_i \mathbf{p_i} \in A_{y_i}] = \int_{A_{y_i}} c e^{-\epsilon ||z - r_i||} dz.$$

By the reverse triangle inequality of Euclidean norm (the difference between the norms of two vectors is at least the norm of the difference between the vectors):

$$-\epsilon ||z - r_i|| = -\epsilon ||z - r_i|| + \epsilon ||z - r'_i|| - $$
$$- \epsilon ||z - r'_i||$$
$$\le \epsilon ||r_i - r'_i|| - \epsilon ||z - r'_i||.$$

Therefore:

$$\Pr[\mathbf{q_i} \in A_{y_i}] \le e^{\epsilon ||r_i - r'_i||} \cdot \int_{A_{y_i}} c e^{-\epsilon ||z - r'_i||} dz,$$

which is $e^{\epsilon ||r_i - r'_i||} \cdot \Pr[\mathbf{q'_i} \in A_{y_i}]$. This completes the first step of the proof.

In the second step, we prove that in the case of Euclidean distance $d_\chi$-STENCIL is $2\epsilon$-$d_\chi$-private. Observe that due to the independence of $\mathbf{p_i}$ the probability that the mechanism outputs $y = (y_1, \ldots, y_N)$ is the product of the probabilities that the $i$-th component of the output is $y_i$ for all $i$. Therefore,

$$\Pr[M(x) = y] = \prod_{i=1}^{N} \Pr[r_i + \mathbf{p_i} \in A_{y_i}]$$
$$\le \prod_{i=1}^{N} e^{\epsilon ||r'_i - r_i||} \Pr[r'_i + \mathbf{p_i} \in A_{y_i}]$$
$$= e^{\epsilon \cdot \sum_{i=1}^{N} ||r'_i - r_i||} \cdot \Pr[M(x') = y].$$

Observe further that due to the definition of $r_i, r_i'$ and to triangle inequality:

$$\left\|\sum_{i=1}^{N} r_i' - r_i\right\| \le \sum_{i=1}^{N}\sum_{j=1}^{N} f_{i,j} \left\|x_i' - x_i\right\|$$

$$= \sum_{j=1}^{N}\sum_{i=1}^{N} f_{i,j} \left\|x_j' - x_j\right\|.$$

But, $\sum_{i=1}^{N} f_{i,j} = C_j$, where $C_j$ is the contribution of the $j$-th token to all other tokens. By Lemma 1, $C_j \le 2$ for all $j$.

Recall that $d_2(x, x') = \sum_{j=1}^{N} \left\|x_j' - x_j\right\|$, and therefore,

$$\Pr[M(x) = y] \le e^{\epsilon \cdot \sum_{i=1}^{N} \left\|r_i' - r_i\right\|} \cdot \Pr[M(x') = y]$$

$$\le e^{2\epsilon \sum_{j=1}^{N} \left\|x_j' - x_j\right\|} \Pr[M(x') = y]$$

$$\le e^{2\epsilon d_2(x,x')} \cdot \Pr[M(x') = y],$$

which completes the proof of the second step.

To prove the third step, we note that the privacy for Euclidean distance holds for any sequence of embeddings $\mathbb{E}(t_i)$ in Euclidean space $\mathbb{R}^\ell$. In particular, it holds if all the embeddings lie on the unit sphere, which would be the case if the vectors are normalized by dividing each vector by its Euclidean norm.

It is well-known that the cosine distance $CD$ between two vectors $x_i, x_i'$ on the unit sphere is equal to $\frac{1}{2}\left\|x_i' - x_i\right\|^2$, for the Euclidean norm $\left\|\cdot\right\|$. On the unit sphere $\left\|x_i' - x_i\right\| \le 2$, which implies that $CD(x_i, x_i') \le \left\|x_i' - x_i\right\|$. Plugging this into the final equations of the previous step we have that

$$\Pr[M(x) = y] \le e^{2\epsilon \sum_{j=1}^{N} \left\|x_j' - x_j\right\|} \Pr[M(x') = y]$$

$$\le e^{2\epsilon \sum_{j=1}^{N} CD(x_j, x_j')} \Pr[M(x') = y]$$

$$\le e^{2\epsilon d_C(x,x')} \cdot \Pr[M(x') = y],$$

which completes the proof.

$\square$

**Remark 1.** *The conclusions of Theorem 1 regarding the privacy of $d_\chi$-STENCIL are pessimistic for large $N$. Lemma 1 proves that the contribution $C_j$ of the $j$-th token is at most 2 in the first and last $L$ tokens, but is exactly 1 otherwise. Repeating the proof of Theorem 1 with this tighter bound on $C_j$, instead of the looser $C_j \le 2$ for all $j$, results in the mechanism providing $2\epsilon$-$d_\chi$ privacy for the two $L$ token substrings at ends of the input, but $\epsilon$-$d_\chi$ privacy for the substring in the middle of the input.*

