# OpenReview forum: "Dchi-Stencil: A Differential Privacy Mechanism for Interacting with LLMs"
_ICLR.cc/2026/Conference — Submitted to ICLR 2026_

### Official Review · Reviewer_W8Bc · 2025-10-31

**Soundness:** 2
**Presentation:** 2
**Contribution:** 2
**Rating:** 4
**Confidence:** 3

**Summary:**

To provide more effective privacy protection during LLM inference, this paper introduces a new method, called 𝑑𝜒-Stencil, by combining existing 𝑑𝜒-privacy mechanism with the STENCIL privacy framework. 𝑑𝜒-Stencil can incorporate more contextual information from the raw user input while injecting noise for privacy protection. The authors also provide sound theoretical privacy analysis of the proposed method to demonstrate the privacy security. Experiment results across four classification-based LLM evaluation benchmark also demonstrate the effectiveness of the proposed method while preserving user privacy.

**Strengths:**

1.	The proposed method is simple yet effective in protecting user privacy during LLM inference. By combining the 𝑑𝜒-privacy mechanism with the STENCIL privacy framework, it preserves more contextual semantics for downstream LLM performance while maintaining strong privacy protection.
2.	The theoretical privacy proof of the proposed method is essential for demonstrating its privacy guarantee. Although the main text lacks a brief explanation, the theoretical analysis provided in the appendix appears sound and well-founded.

**Weaknesses:**

1. Some crucial details are only presented in the Appendix. Although the authors provide theoretical proofs of the privacy mechanism of the proposed 𝑑𝜒-Stencil in Appendix, the main text lacks any brief explanation and merely refers readers to the Appendix (Line 232).
2. Moreover, the relationship among the window size L , the standard deviation σ , and the privacy hyperparameter η in determining the total privacy budget is not clearly explained. While the authors present empirical results in Figures 3 and 4, a theoretical formulation describing how these parameters influence the total privacy budget would strengthen the argument.
3. The evaluation tasks are too simple compared to real-world LLM applications. Current LLMs are typically used for generative tasks such as open-ended question answering or image generation, while the experiments in this paper focus only on classification tasks. Evaluating the proposed methodology on more realistic generative tasks would make the results more convincing.
4. The assumed adversarial attackers appear to be limited. The authors only consider token inversion attacks, however, in real-world scenarios, more advanced attackers may not reconstruct user inputs token by token. Instead, they could feed perturbed text into powerful LLMs and let the LLMs infer the original private content. Considering a wider range of attack techniques would make the evaluation more comprehensive.

**Questions:**

1.	Why the name of the algorithm in Line 219 is STENCIL mechanism rather than 𝑑𝜒-STENCIL mechanism?
2.	As stated in Line 249, the authors use GloVe as the embedding lookup table. However, it is unclear why the raw embedding layers of Flan-T5 and Qwen2.5-1.5B-Instruct are not used directly. Could the use of GloVe lead to inconsistencies in downstream performance?
3.	The hyperparameter setting of 𝑑𝜒 -Stencil in Figure 2 is L=(4,5) and 𝜎 = 0.75. Why the setting of the baseline Stencil is L=9 and 𝜎 = 1.25? Why does the baseline setting keep as your proposed method. More explanation is essitial for make the results in Figure 2 more reliable.

---

### Official Review · Reviewer_1cMy · 2025-10-31

**Soundness:** 2
**Presentation:** 2
**Contribution:** 1
**Rating:** 2
**Confidence:** 3

**Summary:**

This paper proposes $d_{\chi}$-STENCIL, a token-level privacy mechanism privacy-preserving LLM inference combines contextual information in the DP mechanism. Empirical result demonstrates that $d_{\chi}$-STENCIL outperforms or matches recent privacy-preserving baselines in terms of trade-off between utility and privacy compared to existing methods..

**Strengths:**

- The paper combines contextual and semantic information into a DP mechanism.
- The framework has formal privacy guarantee.

**Weaknesses:**

- The contribution of the framework is incremental. It mainly follows the procedure of STENCIL mechanism, with one more step for noise addition. The modification seems trivial.
- The author only evaluates direct token inversion attacks, while more sophisticated attack methods, such as attribute inference attack and adaptive attacks, could be included.

**Questions:**

- Why $d_{\chi}$-STENCIL has better utility than STENCIL in figure 2?

---

### Official Review · Reviewer_AwT2 · 2025-11-01

**Soundness:** 3
**Presentation:** 3
**Contribution:** 3
**Rating:** 6
**Confidence:** 3

**Summary:**

This paper proposes a differential privacy method that incorporates token context into the noise added at a token-level differential privacy application. GloVe embeddings are used to find embeddings for each token in a sequence, then aggregated to get a vector that includes contextual information for a token. Noise is then added to this vector to introduce DP guarantees. The authors find that this is able to produce a higher level of privacy than other techniques that consider semantic similarity to a token, but not the context of the token. They perform a variety of experiments quantifying the effect of different hyperparameters and comparing their method to others, verifying that it outperforms them in privacy.

**Strengths:**

The authors demonstrate the efficacy of their method and perform a number of experiments verifying which parts contribute to this. They compare to other methods and demonstrate that their method is effective. The problem is also relevant and important to the community. The paper is well presented, and the method is clear.

**Weaknesses:**

1. New tokens that do not appear in the embedding model's vocabulary cannot be protected using this method. The authors do not give examples of which tokens these are, which I would like to see, as I am concerned that words not appearing in the vocabulary may be exactly the words that pose higher privacy risks (e.g. names, highly specific vocabulary, newly coined terms specific to some groups, etc.). It would be good to address this point in the paper.

2. The justification for using GloVe embeddings vs another embedding method isn't fully clear to me. It would be good to clarify if this has a performance purpose or if this is for the sake of convenience.

**Questions:**

1. What about using contextual language model embeddings, like those from BERT or more modern models? Is there a benefit to using GloVe embeddings over these?

---

### Meta-Review · Area_Chair_Nd2C · 2026-01-07

**Summary:**

The paper proposes a token-level differential privacy mechanism for interacting with LLMs. Reviewers agree the problem is important. The major concerns include limited novelty compared with STENCIL-style mechanisms, limited threat model and evaluation, and an unclear relationship between parameters.

**Reviewer Concerns:**

No rebuttal was provided. The concerns about limited novelty compared with STENCIL-style mechanisms, limited threat model and evaluation, and an unclear relationship between parameters are still outstanding.

**Reviewer Scores:**

No rebuttal was provided. Reviewers are unlikely to change their scores.

---

### Decision · Program_Chairs · 2026-01-26

Reject